# Multiple NTS neuron populations cumulatively suppress food intake

Weiwei Qiu[1†‡, §], Chelsea R Hutch[2†], Yi Wang[1], Jennifer Wloszek[3], Rachel A Rucker[4], Martin G Myers[1,3,4*], Darleen Sandoval[5*]

[1]Department of Internal Medicine, University of Michigan, Ann Arbor, United States; [2]Department of Surgery, University of Michigan, Ann Arbor, United States; [3]Department of Molecular and Integrative Physiology, University of Michigan, Ann Arbor, United States; [4]Neuroscience Graduate Program, University of Michigan, Ann Arbor, United States; [5]Department of Pediatrics, University of Colorado, Aurora, United States

*For correspondence:
mgmyers@umich.edu (MGM);
darleen.sandoval@cuanschutz.
edu (DS)

†These authors contributed equally to this work

Present address: ‡Zhejiang University-University of Edinburgh Institute, International Campus, Zhejiang University, Hangzhou, China; §Department of Endocrinology, The Second Affiliated Hospital, School of Medicine, Zhejiang University, Hangzhou, China

**Abstract** Several discrete groups of feeding-regulated neurons in the nucleus of the solitary tract (*nucleus tractus solitarius*; NTS) suppress food intake, including avoidance-promoting neurons that express *Cck* (NTS$^{Cck}$ cells) and distinct *Lepr-* and *Calcr*-expressing neurons (NTS$^{Lepr}$ and NTS$^{Calcr}$ cells, respectively) that suppress food intake without promoting avoidance. To test potential synergies among these cell groups, we manipulated multiple NTS cell populations simultaneously. We found that activating multiple sets of NTS neurons (e.g. NTS$^{Lepr}$ plus NTS$^{Calcr}$ [NTS$^{LC}$], or NTS$^{LC}$ plus NTS$^{Cck}$ [NTS$^{LCK}$]) suppressed feeding more robustly than activating single populations. While activating groups of cells that include NTS$^{Cck}$ neurons promoted conditioned taste avoidance (CTA), NTS$^{LC}$ activation produced no CTA despite abrogating feeding. Thus, the ability to promote CTA formation represents a dominant effect but activating multiple non-aversive populations augments the suppression of food intake without provoking avoidance. Furthermore, silencing multiple NTS neuron groups augmented food intake and body weight to a greater extent than silencing single populations, consistent with the notion that each of these NTS neuron populations plays crucial and cumulative roles in the control of energy balance. We found that silencing NTS$^{LCK}$ neurons failed to blunt the weight-loss response to vertical sleeve gastrectomy (VSG) and that feeding activated many non-NTS$^{LCK}$ neurons, however, suggesting that as-yet undefined NTS cell types must make additional contributions to the restraint of feeding.

## Editor's evaluation

This important study describes the role of several populations of NTS neurons in the control of energy balance. The authors provide solid evidence to show that simultaneously stimulating different populations in the NTS induces more potent effects on eating and weight. The work will be of interest to neuroscientists working on neural regulations of energy metabolism.

## Introduction

The ongoing obesity pandemic represents an enormous challenge to human health and longevity (*Centers for Disease Control and Prevention, 2022*). Identifying new therapeutic targets to better combat obesity will require understanding the physiologic systems that modulate feeding and contribute to body weight maintenance. While hypothalamic circuits contribute to the control of feeding and play important roles in maintaining long-term energy balance, many hypothalamic circuits act via the brainstem to suppress food intake (*Cheng et al., 2022*; *Grill, 2014*). Furthermore, the

brainstem receives feeding-related and other information from the gastrointestinal (GI) tract, controls the rhythmic pattern generators that mediate feeding, and can overcome hypothalamically-driven hyperphagia (*Cheng et al., 2022*; *Cheng et al., 2021*).

Many of the most important food intake-controlling brainstem systems lie in the dorsal vagal complex (DVC), which includes the area postrema (AP), the *nucleus tractus solitarius* (NTS), and the dorsal motor nucleus of the vagus (DMV; *Cheng et al., 2022*; *Grill, 2014*). The AP lies outside of the blood-brain barrier and receives circulating signals relevant to feeding, including from several anorexigenic peptides. The NTS, which lies adjacent to and receives input from the AP, also receives input from GI-innervating vagal sensory neurons. Both the AP and NTS project ventrally to the DMV to control GI motility and other physiological functions, in addition to innervating more rostral brain regions to modulate food intake.

Many of these brainstem circuits express receptors for peptides that suppress food intake, including the glucagon-like peptide-1 receptor (GLP1R); the calcitonin receptor (CALCR) and the related amylin receptor (AmyR- a complex of CALCR and a receptor activity modifying protein [RAMP]); and the leptin receptor (LepRb) (*Cheng et al., 2022*; *Ludwig et al., 2021a*; *Cheng et al., 2020a*; *Zhang et al., 2021*). Thus, many of these DVC cell types may contribute to the suppression of feeding by a variety of gut peptide mimetics currently under development or in use for weight loss therapy (*Cheng et al., 2022*).

Recent work from others and us has demonstrated that the AP and NTS neuron types that suppress food intake while mediating avoidance responses associated with gut malaise are distinct from neuron types that promote the non-aversive suppression of food intake (*Cheng et al., 2020a*; *Zhang et al., 2021*; *Roman et al., 2017*; *Palmiter, 2017*; *Campos et al., 2017*; *Cheng et al., 2020b*). Cell types that suppress food intake without causing avoidance include NTS neurons that express *Lepr* (NTS[Lepr] cells) or *Calcr* (NTS[Calcr] cells); in contrast, *Cck*-expressing NTS neurons (NTS[Cck] cells) promote avoidance (*Cheng et al., 2021*; *Cheng et al., 2020a*; *Cheng et al., 2020b*). Each of these cell types represent excitatory (glutamatergic) neurons *Ludwig et al., 2021a*; these neurons also contain the neuropeptides glucagon-like peptide-1 (GLP-1; NTS[Lepr] neurons), prolactin releasing peptide (PrRP; NTS[Calcr] neurons), and cholecystokinin (CCK; NTS[Cck] neurons) (*Cheng et al., 2021*; *Ludwig et al., 2021a*; *Huo et al., 2008*; *Cheng et al., 2020b*).

Bariatric surgeries, including vertical sleeve gastrectomy (VSG), decrease meal size and overall food intake to promote dramatic weight loss in many patients (*Peterli et al., 2018*), as well as in animal models of VSG (*Stefater et al., 2010*). The irreversibility and side effects of the procedure, together with the inability to perform enough surgeries to treat the number of patients in need, prevents the use of this intervention at a population scale, however (*Altieri et al., 2021*). Hence, we must understand the mechanisms by which VSG mediates its effects to help identify therapeutic targets with which to medically mimic its actions.

Hypothesizing that the effects of distinct NTS neuron types are cumulative, we simultaneously manipulated (activation or silencing) multiple NTS cell types to determine their roles in energy balance and the response to VSG.

## Results

### Augmentation of food intake suppression by coordinated activation of NTS[Lepr] and NTS[Calcr] neurons

We have previously shown that NTS[Lepr] and NTS[Calcr] cells represent distinct populations of NTS neurons (*Ludwig et al., 2021a*; *Cheng et al., 2020a*), and that the activation of either population mediates the non-aversive suppression of food intake (*Cheng et al., 2020a*; *Cheng et al., 2020b*). To determine the relative ability of each of these cell types to mediate the suppression of food intake and to determine whether they might play cumulative roles in the suppression of feeding, we injected an adeno-associated virus (AAV) to cre-dependently express the activating (hM3Dq) designer receptor exclusively activated by designer drugs (DREADD; AAV[FLEX-hM3Dq] *Sternson and Roth, 2014*; *Zhu and Roth, 2014*) into the NTS of *Lepr[Cre]*, *Calcr[Cre]*, or dual *Lepr[Cre];Calcr[Cre]* mice (producing Lepr[Dq], Calcr[Dq], and LC[Dq] mice, respectively). We then examined the effects of CNO-mediated hM3Dq activation on food intake and related parameters in these mice (*Figure 1*). As expected, CNO treatment promoted

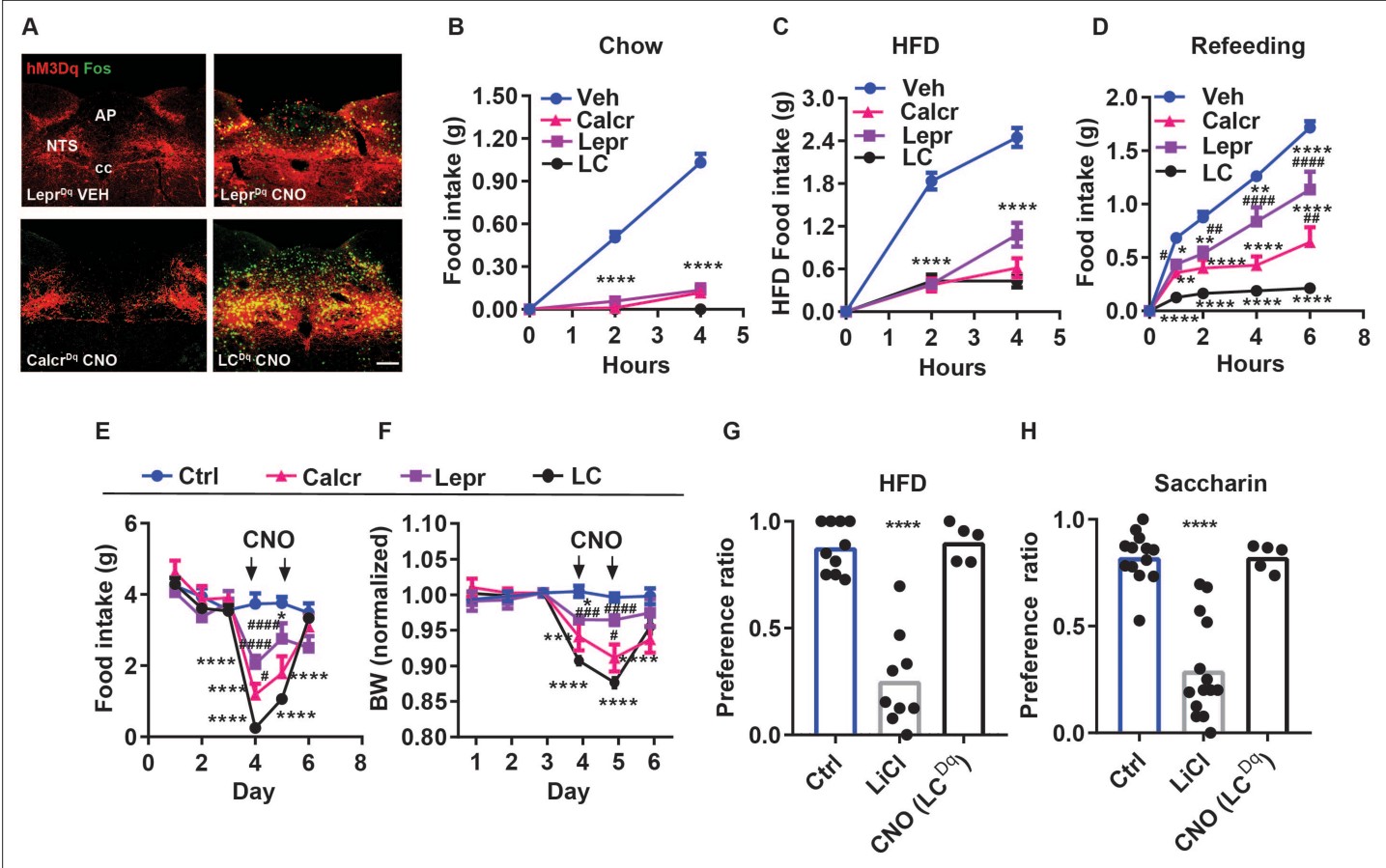

**Figure 1.** Additive suppression of food intake with combined activation of NTS[Calcr] and NTS[Lepr] neurons. (**A**) Representative images showing mCherry-IR (hM3Dq, red) and FOS-IR (green) in the NTS (approximately Bregma –7.56 mm) of Lepr[Dq], Calcr[Dq] and LC[Dq] mice following treatment with saline (Lepr[Dq] VEH) or CNO (Lepr[Dq] CNO, Calcr[Dq] CNO and LC[Dq] CNO IP, 1 mg/kg) for 2 hr before perfusion. NTS: nucleus of the solitary tract, AP: area postrema, cc: central canal. All images were taken at the same magnification; scale bar equals 150 μm. (**B–D**) Food intake in chow-fed Lepr[Dq] (Lepr), Calcr[Dq] (Calcr) and LC[Dq] (LC) mice over the first 4 hr of the dark phase following vehicle (Veh) or CNO injection (IP, 1 mg/kg) when provided with chow (**B**, n=25 for Veh group; n=20 for both Lepr and Calcr groups; n=8 for LC group) or HFD (**C**, n=21 for Veh group; n=6 for Lepr group; n=7 for Calcr group; n=8 for LC group). (**D**) Food intake for the same groups of mice over the first 6 hours of refeeding during the light cycle following an overnight fast (**D**, n=23 for Veh group; n=8 for Lepr group; n=7 for Calcr group; n=7 for LC group) following with CNO (IP, 1 mg/kg) or vehicle (Veh). (**E–F**) Control (Ctrl; AAV[GFP] or AAV[cre-GFP]-injected, n=5) or Lepr[Dq] (Lepr, n=5), Calcr[Dq] (Calcr, n=5) and LC[Dq] (LC, n=8) mice were treated with vehicle for three baseline days, followed by 2 days of twice daily treatment with CNO (IP 1 mg/kg), followed by two additional days of Veh treatment. Daily food intake (**E**) and body weight (shown as change from baseline) (**F**). Vehicle and CNO treatment are denoted on the graphs. (**G, H**) CTA assays: Mice were treated with vehicle (Veh, n=10), LiCl (IP, 126 mg/kg, n=9), or CNO (IP, 1 mg/kg, n=5) during exposure to a novel tastant (HFD) (**G**) or saccharin (**H**) paired with vehicle (Veh, n=13), LiCl (IP, 126 mg/kg, n=14), or CNO (IP, 1 mg/kg, n=5). Shown is mean +/-SEM; Two-way ANOVA, sidak's multiple comparisons test was used; *p<0.05, **p<0.01, ***p<0.001, ****p<0.0001 vs vehicle or Ctrl; #p<0.05, ##p<0.01, ###p<0.001, ####p<0.0001 vs LC.

The online version of this article includes the following source data and figure supplement(s) for figure 1:

**Source data 1.** Data for *Figure 1* panels B-H.

**Figure supplement 1.** Inhibiting NTS[LC] neurons increases food intake.

**Figure supplement 1—source data 1.** Source data for *Figure 1—figure supplement 1* panels B-E.

FOS-immunoreactivity (-IR) in the NTS of Lepr[Dq], Calcr[Dq] and LC[Dq] mice, with the LC[Dq] mice displaying increased FOS-IR compared to animals in which we had transduced only one population (*Figure 1A*). As we showed previously (*Cheng et al., 2020a*; *Cheng et al., 2020b*), CNO almost completely abrogated food intake at the onset of the dark cycle in chow-fed Lepr[Dq] and Calcr[Dq] mice; LC[Dq] mice responded similarly (*Figure 1B*). The same was true for mice fed high-fat diet (HFD) at the onset at the dark cycle, although the activation of NTS[Lepr] neurons tended to suppress feeding less than NTS[Calcr] and NTS[LC] neurons at the 4 hr time point in this assay (*Figure 1C*). Following an overnight fast, the activation of NTS[Lepr] neurons significantly decreased refeeding over 6 hr; the activation of NTS[Calcr]

cells alone decreased refeeding further than NTS^Lepr neurons, and the activation of NTS^LC neurons decreased refeeding even further- by approximately 90% (*Figure 1D*).

Similarly, activating NTS^Lepr neurons decreased feeding and body weight over a two-day treatment, activating NTS^Calcr neurons suppressed these parameters more effectively, and activating both cell types in LC^Dq mice reduced feeding by >90% for the first 24 hr and continued to reduce food intake compared to the individual cell types during the second 24 hr of the experiment (*Figure 1E and F*). While we had intended to continue treatment for another day, we discontinued CNO treatment after 48 hr because of animal welfare concerns due to the amount of weight loss exhibited by the LC^Dq mice. Hence, NTS^Calcr neurons suppress feeding more robustly than NTS^Lepr neurons, and NTS^Calcr and NTS^Lepr neurons suppress feeding cumulatively.

To determine whether the dramatic decrease in feeding promoted by NTS^LC neuron activation provoked avoidance responses, we examined whether CNO treatment in LC^Dq mice would provoke a CTA. We found that, although the gut malaise associated with LiCl injection promoted a robust CTA to either the novel exposure to HFD or saccharine in drinking water, NTS^LC activation failed to provoke such a response (*Figure 1G and H*). Thus, the coordinated activation of two non-aversive cell types (NTS^Lepr and NTS^Calcr *Cheng et al., 2020a*; *Cheng et al., 2020b*; *Gaykema et al., 2017*) fails to promote a CTA despite almost abolishing food intake.

We also expressed the inhibitory (hM4Di) DREADD in NTS^LC cells to examine the effect of acutely inhibiting these cells on short-term food intake (*Figure 1—figure supplement 1*). While CNO failed to significantly increase food intake during the first four hours of the dark cycle, stimulating hM4Di in NTS^LC cells increased food intake during refeeding with chow or HFD following an overnight fast. Thus, NTS^LC neurons contribute to the restraint of food intake during refeeding.

We previously showed that the long-term tetanus toxin (TetTox)-mediated silencing of NTS^Calcr neurons promotes increased food intake and body weight in animals exposed to HFD (*Cheng et al., 2020a*). To determine whether silencing NTS^Lepr cells might mediate similar effects (thereby implicating this additional NTS cell type in the control of long-term energy balance), we injected an AAV to cre-dependently express TetTox into the NTS of *Lepr^Cre* mice (AAV^FLEX-TetTox *Kim et al., 2009*; Lepr^TetTox mice) and examined their food intake and body weight during 6 weeks of chow feeding followed by an additional 6 weeks of HFD feeding (*Figure 2A–D*). While we detected no significant increase in food intake or body weight during chow feeding for Lepr^TetTox mice, these animals displayed a modest increase in food intake on HFD and gained more body weight over 6 weeks than control animals. Thus, NTS^Lepr neurons (like NTS^Calcr cells) contribute to the long-term control of energy balance.

To determine whether the additional silencing of NTS^Calcr neurons might exacerbate the increased food intake and body weight displayed by Lepr^TetTox mice, we used AAV^FLEX-TetTox to cre-dependently express TetTox in the NTS of *Lepr^Cre;Calcr^Cre* animals (LC^TetTox mice) (*Figure 2E–H*). While we detected no significant increase in food intake or body weight during 6 weeks of chow feeding in LC^TetTox mice compared to controls, LC^TetTox mice ate significantly more food and gained (~10 grams) more body weight than controls when exposed to HFD for 6 weeks.

Continuous analysis of feeding demonstrated increased food intake by LC^TetTox mice in the dark (active) cycle, consistent with decreased suppression of food intake in response to normal feeding (*Figure 2—figure supplement 1*). While comparing results among different strains of mice studied at different times requires cautious interpretation, these findings suggest that the silencing of multiple NTS neuron populations (as in LC^TetTox mice) increases feeding and weight gain relative to silencing single populations (NTS^Lepr in the current study, as well as previously-studied NTS^Calcr cells previously *Cheng et al., 2020a*; *Figure 2I*).

Given the dramatic effects on food intake and body weight observed upon silencing NTS^LC cells, we decided to test the requirement for the function of these cells in the weight loss response to VSG. We thus generated HFD-fed control and LC^TetTox animals as in *Figure 2* and subjected them to VSG (*Figure 2—figure supplement 1*). While, as expected, HFD-fed LC^TetTox mice weighed more than HFD-fed control animals at the time of surgery, VSG also promoted more weight loss in these animals than in controls (both in terms of total weight lost and percent body weight lost) and similarly improved measures of glycemic control. Thus, NTS^LC neurons are not required for VSG-mediated weight loss, and other cell types must contribute to the suppression of food intake following VSG.

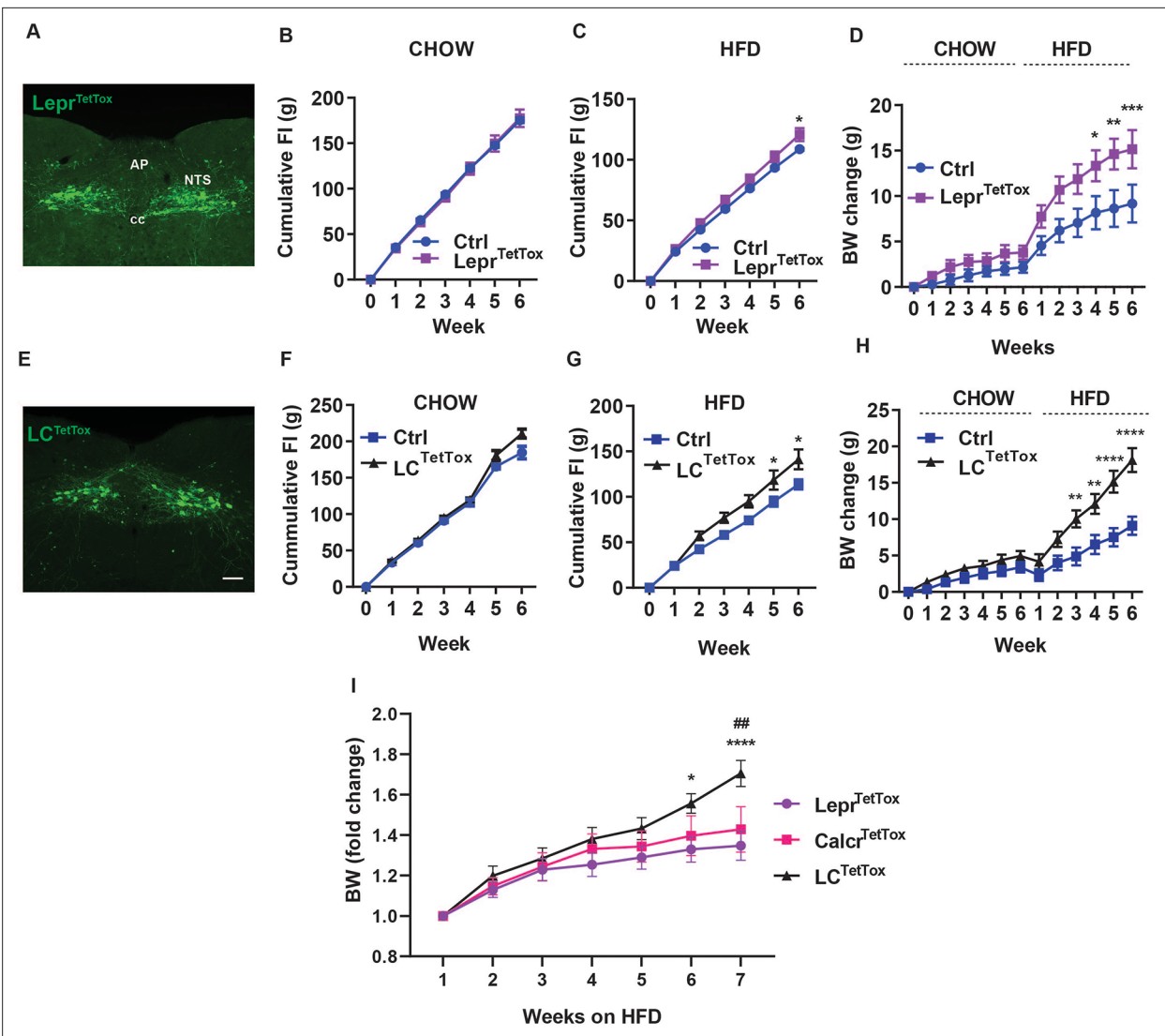

**Figure 2.** Silencing NTS[Lepr] neurons or NTS[LCS] neurons exacerbates diet-induced obesity (DIO). Representative images showing GFP-IR (green) from in Lepr[TetTox] (A) and LC[TetTox] (E) mice (approximately Bregma –7.56 mm). All images were taken at the same magnification; scale bar equals 150 μm. NTS: nucleus of the solitary tract, AP: area postrema, cc: central canal. B-D, F-H show cumulative food intake during chow (B, F) and HFD feeding (C, G) and body weight (change from baseline) (D, H) for control (Ctrl; AAV[FLEX-hM3Dq]-injected), Lepr[TetTox] (B–D) and LC[TetTox] (F–H) mice following VSG, during which time they were fed with chow for 5–6 weeks and HFD for an additional 6 weeks. n=7 each group for B and C, n=8 each group for D for Ctrl and Lepr[TetTox] group; n=7 for Ctrl and n=8 for LC[TetTox] group in F and G, n=11 for Ctrl and n=12 for LC[TetTox] group in H. (I) Shows weight gain from the onset of HFD feeding for Lepr[TetTox] (from D) and LC[TetTox] mice (from H) in comparison to previously-published Calcr[TetTox] mice (*Cheng et al., 2020a*). Shown are mean +/-SEM. Two-way ANOVA, sidak's multiple comparisons test was used; *p<0.05, **p<0.01, ***p<0.001, ****p<0.0001 vs Ctrl, except in I, where these refer to the comparison between Lepr[TetTox] and LC[TetTox]; ##p<0.01 between Calcr[TetTox] and LC[TetTox].

The online version of this article includes the following source data and figure supplement(s) for figure 2:

**Source data 1.** Source data for *Figure 2* Panels B-D, F-I.

**Figure supplement 1.** Silencing NTS[LC] neurons increases food intake and body weight but fails to abrogate the weight loss effects of VSG.

**Figure supplement 1—source data 1.** Source data for *Figure 2—figure supplement 1*, Panels A-L.

## Cumulative roles for avoidance and non-avoidance-provoking NTS cell types for the control of energy balance

While NTS[Lepr] and NTS[Calcr] cells both mediate the non-aversive suppression of food intake, activation of NTS[Cck] neurons promotes avoidance (e.g. CTA) responses, as well as decreasing feeding (*Cheng et al., 2020a*; *Roman et al., 2017*; *Cheng et al., 2020b*; *Roman et al., 2016*). Thus, to understand

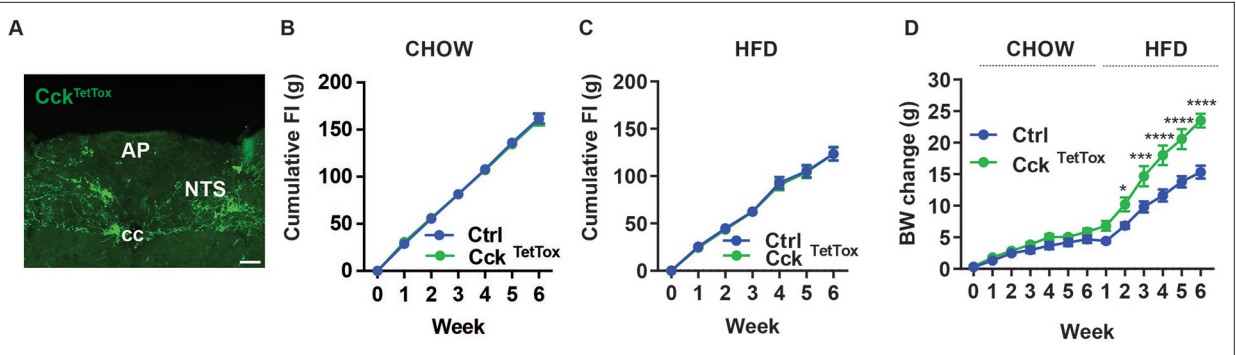

**Figure 3.** Silencing NTS[Cck] neurons exacerbates DIO. (**A**) Representative images showing GFP-IR (green) in the NTS of Cck[TetTox] mice (approximately Bregma –7.56 mm). Images were taken at the same magnification; scale bar equals 150 µm. NTS: nucleus of the solitary tract, AP: area postrema, cc: central canal. (**B–D**) Cumulative food intake during chow (**B**) and HFD (**C**) feeding and body weight (**D**, change from baseline) is shown for control (Ctrl; AAV[FLEX-hM3Dq]-injected) and Cck[TetTox] mice following surgery, during which time they were fed with chow for 6 weeks and the HFD for an additional 6 weeks. n=5 each group for **B**, n=6 for Ctrl and n=7 for Cck[Tettox] group in **C**, n=11 for Ctrl group and n=10 for Cck[TetTox] group in **D**. Shown is mean +/- SEM. Two-way ANOVA, sidak's multiple comparisons test was used, *p<0.05, ****p<0.0001 vs Ctrl.

The online version of this article includes the following source data and figure supplement(s) for figure 3:

**Source data 1.** Source Data for *Figure 3*, panels B-D.

**Figure supplement 1.** Response of Cck[TetTox] mice to VSG.

**Figure supplement 1—source data 1.** Source data for *Figure 3—figure supplement 1*, panels A-I.

potential roles for avoidance-provoking NTS[Cck] cells in the control of energy balance, we injected AAV[FLEX-TetTox] to mediate the cre-dependent expression of tetanus toxin (TetTox) into the NTS of *Cck^Cre* mice (Cck[TetTox] mice; *Figure 3A*) and examined their food intake and body weight during 6 weeks of chow feeding followed by an additional 6 weeks of HFD feeding (*Figure 3B–D*). While we detected no significant increase in food intake or body weight during chow feeding for Cck[TetTox] mice, these animals gained more body weight over 6 weeks than control animals during HFD feeding. Thus, while it is possible that the inaccuracies inherent to food intake measurement prevented us from detecting an alteration in food intake in the Cck[TetTox] animals, silencing NTS[Cck] neurons might impact body weight by other means, including by altering calorie absorption or energy expenditure. In any case, NTS[Cck] neurons, like NTS[Calcr] and NTS[Lepr] cells, contribute to the long-term control of energy balance.

To determine whether avoidance-provoking NTS[Cck] neurons might play a role in the weight loss response to VSG, we decided to test whether the function of these cells for the weight loss response to VSG. We thus generated HFD-fed control and Cck[TetTox] animals as in *Figure 3* and subjected them to VSG (*Figure 3—figure supplement 1*). While, as expected, HFD-fed Cck[TetTox] mice weighed more than HFD-fed control animals at the time of surgery, VSG promoted similar weight loss in these animals as in controls.

To understand potential cumulative effects on food intake for avoidance-provoking and non-aversive NTS cell types, we chose to examine the effects of manipulating NTS[Cck] neurons (*Cheng et al., 2020a*; *Roman et al., 2017*; *Roman et al., 2016*) in combination with NTS[LC] cells (collectively, NTS[LCK] cells; *Figures 4–5*). We began by injecting AAV[FLEX-hM3Dq] (*Sternson and Roth, 2014*; *Zhu and Roth, 2014*) into the NTS of *Lepr^Cre;Calcr^Cre;Cck^Cre* mice (producing LCK[Dq] mice; *Figure 4*). We then examined the effects of CNO-mediated activation of NTS[LCK] cells on food intake and related parameters in these animals.

As expected, CNO treatment promoted FOS-IR in the NTS of LCK[Dq] mice (*Figure 4A*). As for LC[Dq] mice (*Figure 1*), CNO almost completely abrogated food intake at the onset of the dark cycle in chow-fed and HFD-fed LCK[Dq] mice, and strongly suppressed refeeding following an overnight fast (*Figure 4B–D*). We also assessed the effect of activating NTS[LCK] neurons over a two-day treatment. CNO treatment dramatically decreased food intake in LCK[Dq] mice and resulted in even larger decreases in body weight that in LC[Dq] mice (*Figures 1E–F and 4E–F*), confirming the effectiveness of the coordinated action of NTS[LCK] neurons for the control of food intake and body weight.

Because food intake during CNO treatment was similarly low for LC[Dq] and LCK[Dq] mice (it is not possible to decrease food intake below zero), while the LCK[Dq] mice lost more body weight, it is

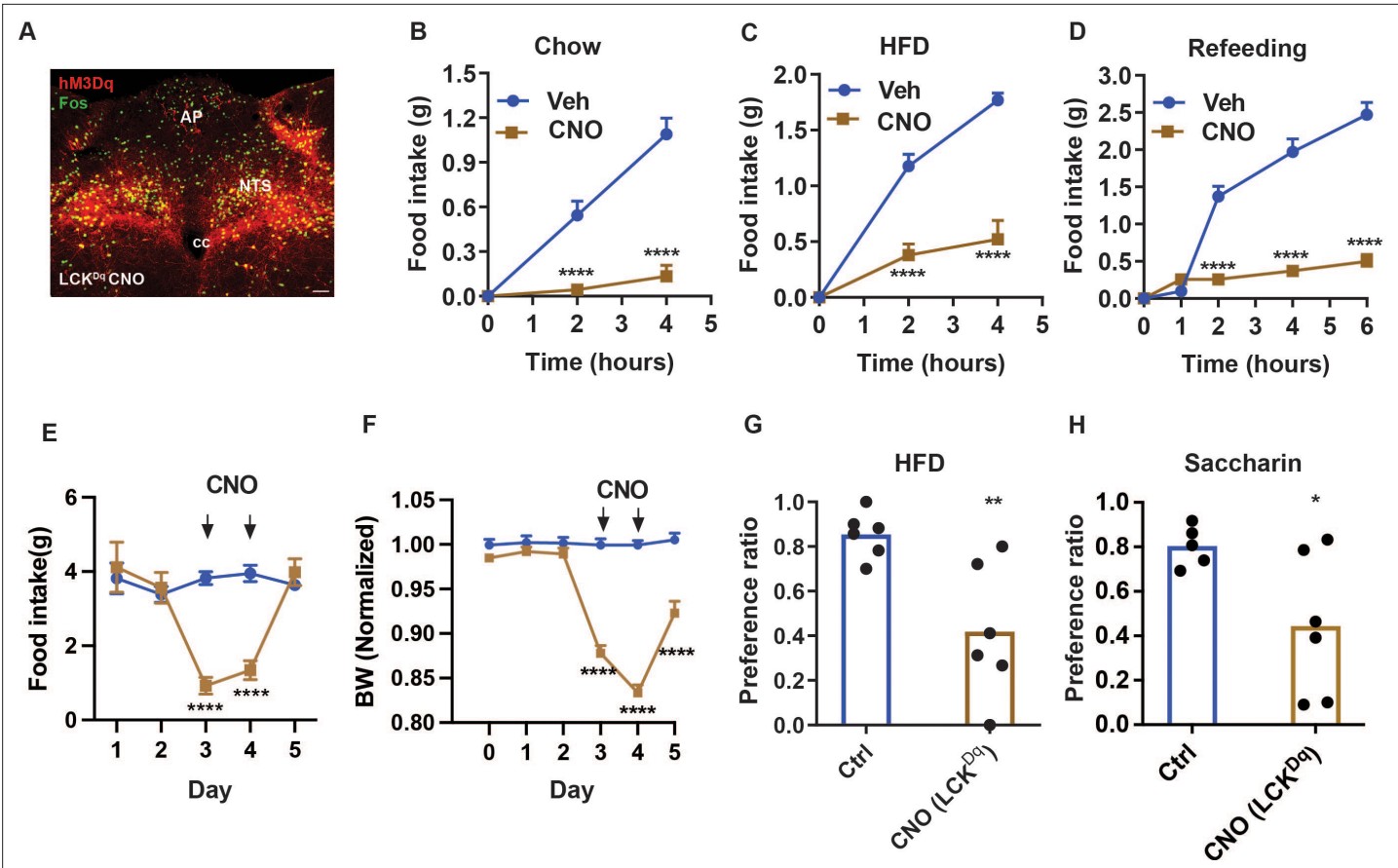

**Figure 4.** NTS[LCK] neuron activation decreases food intake and promotes a CTA. (**A**) Representative images showing mCherry-IR (hM3Dq, red) and FOS-IR (green) in the NTS (approximately Bregma –7.56 mm) of LCK[Dq] mice following treatment with CNO (IP, 1 mg/kg) 2 hr before perfusion. Scale bar equals 150 µm. NTS: nucleus of the solitary tract, AP: area postrema, cc: central canal. (**B–D**) Food intake in LCK[Dq] mice over the first 4 hr of the dark phase during expose to chow (**B**) or HFD (**C**) (n=9 per group), and during the first 6 hr of refeeding in the light cycle following an overnight fast (**D**, n=7 per group) during treatment with CNO (IP, 1 mg/kg) or vehicle. (**E–F**) Control (Ctrl; AAV[GFP]-injected, n=7) or LCK[Dq] mice (n=7) mice were treated vehicle (Veh) for 1 days, followed by 2 days with CNO (1 mg/kg, IP, twice per day) and daily food intake (**E**) and body weight (change from baseline) (**F**) were determined. Veh and CNO treatment are denoted on the graphs. (**G, H**) Ctrl (uninjected littermates) or LCK[Dq] mice were treated with Veh, LiCl, or CNO (IP, 1 mg/kg) during exposure to a novel tastant (HFD (**G**) or saccharin (**H**)); n=6 per group. Shown is mean +/-SEM; Two-way ANOVA, sidak's multiple comparisons test was used in all figures, except panels G and H where unpaired T test was used; *p<0.05, **p<0.01, ***p<0.001, ****p<0.0001 vs Veh or Ctrl.

The online version of this article includes the following source data for figure 4:

**Source data 1.** Source data for *Figure 4*, Panels B-H.

possible that the activation of NTS[Cck] neurons promotes negative energy balance by altering energy absorption or expenditure as well as by decreasing feeding. Indeed, silencing NTS[Cck] cells increased body weight without measurably altering food intake (*Figure 3*). We also found that, unlike NTS[LC] cells, activating NTS[LCK] cells provoked a strong CTA to HFD or saccharine in water, consistent with the avoidance-provoking nature of NTS[Cck] neurons and the dominant nature of avoidance-provoking over non-avoidance-provoking hindbrain signals (*Figure 4G–H*).

To determine whether silencing NTS[Cck] neurons might exacerbate the increased food intake and body weight displayed by LC[TetTox] mice (and vice-versa), we injected AAV[FLEX-TetTox] into the NTS of *Lepr-Cre;Calcr-Cre;Cck-Cre* animals (LCK[TetTox] mice) (*Figure 5A*). While we had not detected increased food intake on chow diet in Lepr[TetTox], Calcr[TetTox], Cck[TetTox], or LC[TetTox] animals (*Figures 2–3*), LCK[TetTox] mice displayed a significant increase in food intake on chow, as well as on HFD (*Figure 5B–C*). Although LCK[TetTox] mice also tended to gain more weight on chow diet, the effect was not statistically significant. However, these animals became significantly heavier than controls after the first week on HFD, and continued to gain weight relative to controls with each additional week (*Figure 5D*).

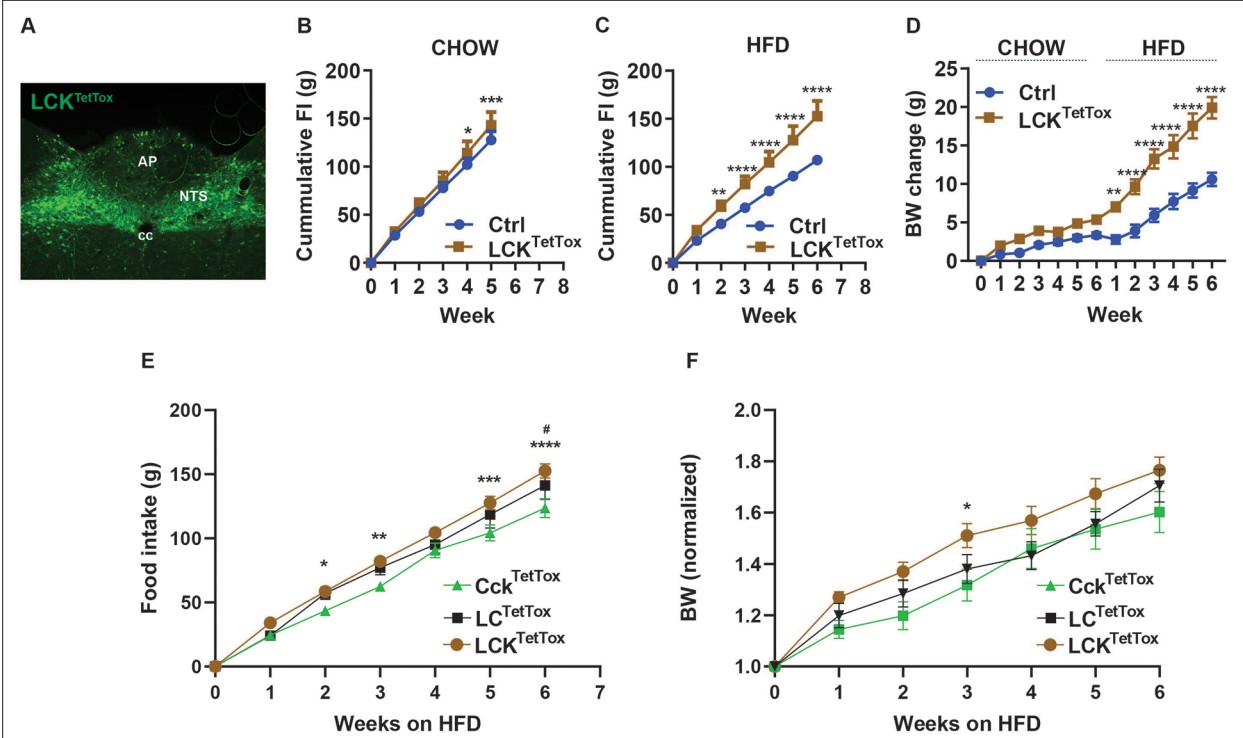

**Figure 5.** Silencing of NTS^LCK neurons increases food intake and body weight. (**A**) Representative image showing GFP-IR (green) in the NTS of LCK^TetTox mice (approximately Bregma –7.56 mm). Scale bar equals 150 μm. NTS: the nucleus of the solitary tract, AP: area postrema, cc: central canal. (**B–D**) Cumulative food intake in response to chow (**B**, n=8 per group) and HFD (**C**, n=5 for control (Ctrl; AAV^FLEX-hM3Dq-injected) group and n=8 for LCK^TetTox group) and body weight (**D**, change from baseline; n=6 for Ctrl group and n=8 for LCK^TetTox group) is shown for Ctrl and LCK^TetTox mice following surgery, during which time they were fed chow for 6 weeks and HFD for an additional 6 weeks. (**E–F**) Comparisons of cumulative HFD food intake (**E**) and body weight (normalized to baseline) for Cck^TetTox (from **Figure 3C**), LC^TetTox (from **Figure 2G**), and LCK^TetTox (from panel **C**) mice. Shown is mean +/-SEM; Two-way ANOVA, sidak's multiple comparisons test was used; *p<0.05, **p<0.01, ***p<0.001, ****p<0.0001 vs Ctrl except **E** and **F**, for which these indicate differences between Cck^TetTox and LCK^TetTox; #p<0.05 for Cck^TetTox vs LC^TetTox.

The online version of this article includes the following source data for figure 5:

**Source data 1.** Source data for **Figure 5**, panels B-F.

While one must exercise caution when comparing results among different strains of mice, we observed increased HFD food intake for LCK^TetTox animals compared to Cck^TetTox and LC^TetTox mice (**Figure 5E**). Similarly, LCK^TetTox mice tended to gain more weight on HFD compared to LC^TetTox and Cck^TetTox animals (**Figure 5F**). These findings suggest that simultaneously silencing multiple food intake controlling NTS neuron populations cumulatively increases food intake and body weight, irrespective of whether or not the neurons mediate avoidance responses.

To better understand the potential mechanisms underlying the cumulative nature of signals from the various NTS cell types, we treated control (wild-type), Lepr^Dq, Calcr^Dq, Cck^Dq, LC^Dq, and LCK^Dq mice with CNO and examined FOS-IR in the AP, NTS, parabrachial nucleus (PBN), and dorsomedial hypothalamic nucleus (DMH) (**Figure 6—figure supplement 1**). In addition to revealing the expected increase in NTS FOS-IR with the activation of each additional cell type (**Figure 6A and D**), we also observed differences in the magnitude and distribution of FOS-IR between cell types in other areas in a manner that was roughly additive. While CNO provoked little AP FOS-IR in Lepr^Dq mice, the activation of NTS^Calcr or NTS^Cck neurons tended to increase AP FOS-IR (**Figure 6A and C**). Furthermore, while CNO treatment in LC^Dq mice did not increase AP FOS-IR compared to that observed in Calcr^Dq mice, LCK^Dq demonstrated the greatest amount of AP FOS-IR.

In the PBN (**Figure 6B and D**), the LC^Dq mice demonstrated increased FOS-IR compared to Lepr^Dq or Calcr^Dq alone, although neither of these induced the colocalization of FOS-IR with CGRP (the activation of which is known to promote avoidance behavior **Carter et al., 2015**). In contrast, the activation of NTS^Cck cells alone or in combination with the other cell types provoked FOS/CGRP colocalization,

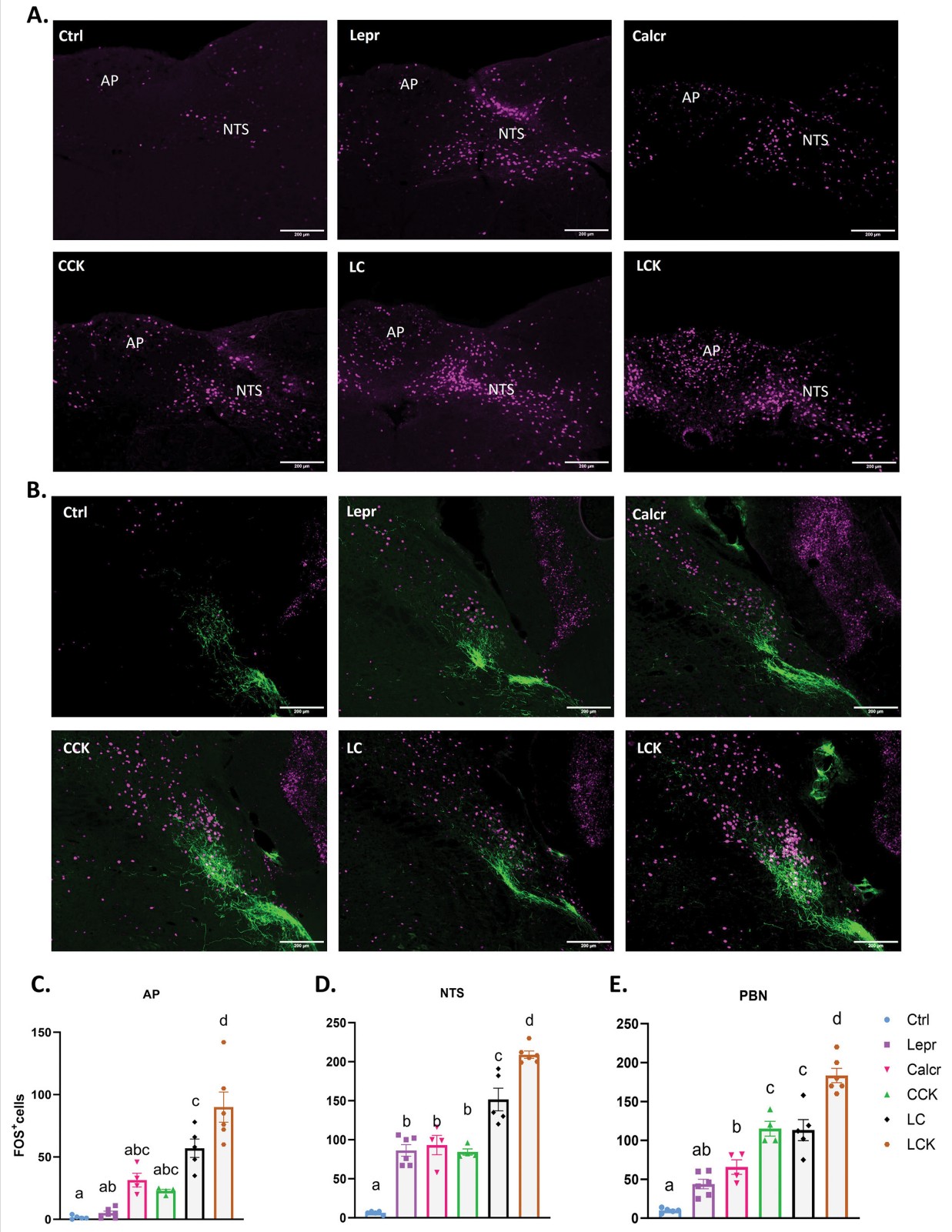

**Figure 6.** FOS-IR in the AP, NTS, and PBN following activation of single or multiple populations of NTS neurons. Control (Ctrl; AAV$^{GFP}$-injected), Lepr$^{Dq}$, Calcr$^{Dq}$, CCK$^{Dq}$, LC$^{Dq}$, and LCK$^{Dq}$ animals were fasted overnight, injected with CNO (1 mg/kg) and perfused 2 hr later. Brains were processed and stained for FOS (or FOS and CGRP)-IR. (**A**) Representative images of the AP and NTS from the indicated animal type showing FOS-IR (magenta) (approximately Bregma –7.56 mm). (**B**) Representative images of the PBN (approximately Bregma –5.20 mm) from the indicated animal type showing FOS-IR (magenta)

*Figure 6 continued on next page*

*Figure 6 continued*

and CGRP-IR (green). For **A-B** scale bars represent 200 μm. (**C-E**) Quantification of FOS-IR in the indicated regions for CNO-treated animals of each type. All bars, mean +/-SEM is shown, along with the distribution of values for each bar. Calcr$^{Dq}$, Cck$^{Dq}$, n=4 each; Ctrl, LC$^{Dq}$, n=5 each; Lepr$^{Dq}$, LCK$^{Dq}$, n=6 each. Bars with the same letter are not different by ANOVA (i.e. bars without overlapping letters are different by ANOVA).

The online version of this article includes the following source data and figure supplement(s) for figure 6:

**Source data 1.** Source data for *Figure 6*, panels C-E.

**Figure supplement 1.** FOS-IR in the DMH following activation of single or multiple populations of NTS neurons.

**Figure supplement 2.** Appropriate response to VSG for LCK$^{TetTox}$ mice.

**Figure supplement 2—source data 1.** Source data for *Figure 6—figure supplement 2*, panels A-L.

**Figure supplement 3.** Activation of NTS$^{LCK}$ and other NTS neurons in response to refeeding and VSG.

**Figure supplement 3—source data 1.** Source Data for *Figure 6—figure supplement 3*, panels B-C, E-F.

consistent with the avoidance response provoked by any stimulus that includes the activation of NTS$^{Cck}$ cells. CNO treatment in LCK$^{Dq}$ mice provoked increased PBN FOS-IR compared to LC$^{Dq}$ mice, suggesting the existence of substantial numbers of distinct PBN targets for NTS$^{Cck}$ compared to NTS$^{LC}$ neurons. The activation of DMH FOS-IR similarly increased with the stimulation of additional NTS cell types (*Figure 6—figure supplement 1*). Together, these findings suggest that each NTS population targets at least partially distinct downstream neurons in several brain regions. Thus, the activation of additional downstream targets by the stimulation of more populations of NTS neurons likely underlies the cumulative effects of activating multiple NTS cell types.

## Evidence for the presence of additional NTS cell types that may mediate additional cumulative effects on food intake

Given the dramatic effects on food intake and body weight observed upon silencing NTS$^{LCK}$ cells, we decided to test the requirement for the combined function of these cells for the weight loss response to VSG (*Figure 6—figure supplement 2*). While, as expected, HFD-fed LCK$^{TetTox}$ mice weighed more than HFD-fed control animals at the time of surgery, VSG promoted more body and fat mass loss in these animals than in controls. Food intake was decreased by VSG in LCK$^{TetTox}$ animals, although it remained higher than for control mice that had received VSG, presumably reflecting the increased food intake of these mice at baseline. Thus, VSG normalized body weight and fat mass in LCK$^{TetTox}$ mice and promoted similar improvements in indices of glucose homeostasis in LCK$^{TetTox}$ and control mice. These findings suggest that, despite the important roles for these cells in the physiologic restraint of feeding and body weight, neurons other than NTS$^{LCK}$ cells must mediate the response to VSG.

To determine whether additional (non-LCK) NTS neurons might contribute to the refeeding response in control animals and/or in mice subjected to VSG, we quantified the FOS response to refeeding in control or VSG animals expressing reporter fluorophores in NTS$^{LCK}$ cells (*Figure 6—figure supplement 3*). This analysis revealed that NTS$^{LCK}$ cells represent approximately 20% of feeding-activated NTS neurons in control mice, and approximately 35% of such neurons following VSG. Thus, in addition to neurons in the AP, hypothalamus, and elsewhere, feeding stimulates non-LCK neurons in the NTS and these feeding-activated non-LCK cells presumably contribute to food intake suppression and body weight control.

## Discussion

Our findings reveal that distinct groups of food intake-inhibiting NTS neurons act cumulatively to suppress food intake under physiologic feeding as well as during artificial activation. We also show that the enhanced suppression of food intake mediated by the activation of multiple populations of individually non-aversive NTS neurons does not promote avoidance responses irrespective of the strength of food intake suppression, consistent with the notion that the pathways that promote the non-aversive suppression of food intake are distinct from those that promote avoidance responses. Finally, although NTS$^{Lepr}$, NTS$^{Calcr}$, and NTS$^{Cck}$ neurons individually and coordinately restrain body weight gain during HFD feeding but these cells are dispensable for weight loss in response to VSG and feeding activates additional NTS cell types, suggesting roles for additional neuron types (including within the NTS) in feeding suppression.

*Calcr* and *Lepr* expression do not colocalize in the same NTS neurons and unbiased clustering methods reveal that NTS$^{Calcr}$ and NTS$^{Lepr}$ neurons map to distinct populations of NTS neurons in mice (NTS$^{Calcr}$ neurons map to GLU11, while NTS$^{Lepr}$ neurons map mainly to GLU13 and GLU2; GLU13 controls feeding) (*Ludwig et al., 2021a*; *Cheng et al., 2020a*; *Cheng et al., 2020b*; *Gaykema et al., 2017*). Each of these populations strongly suppresses food intake but fails to promote a CTA when activated, suggesting that these NTS cells activate pathways that blunt feeding without engaging distinct avoidance pathways (*Cheng et al., 2021*; *Cheng et al., 2020a*; *Cheng et al., 2020b*; *Gaykema et al., 2017*; *Tsang et al., 2020*). Our finding that the combined activation of these two cell types reduces feeding by >90% over the first 24 hr of treatment, but fails to promote a CTA, further reinforces the notion that brainstem pathways need not provoke aversive responses to suppress food intake.

The magnitude of the anorexia that accompanies the combined activation of NTS$^{Calcr}$ and NTS$^{Lepr}$ cells relative to the response to the activation of a single population also reveals that distinct populations of food intake-suppressing NTS neurons can act cumulatively to suppress feeding more effectively. Similarly, while the silencing of either of these populations can increase physiologic food intake and body weight in HFD-fed animals, silencing multiple populations provokes greater feeding and body weight gain. Hence, the effects of the various sets of food intake-suppressing NTS neurons must augment each other to suppress physiologic feeding, at least under HFD-fed conditions.

Examining the accumulation of FOS-IR in the AP, NTS and known projection targets of the NTS (PBN and DMH) demonstrates the cumulative activation of downstream neurons following the stimulation of multiple NTS neuron populations. Indeed, the distribution of evoked FOS-IR varies somewhat between NTS cell types; unlike NTS$^{Calcr}$ and NTS$^{Cck}$ cells, NTS$^{Lepr}$ neurons poorly activate AP cells, while NTS$^{Cck}$ cells (but not NTS$^{Calcr}$ and/or NTS$^{Lepr}$ cells) activate PBN CGRP neurons. These findings thus suggest that each NTS neuron population activates an at least partially distinct set of downstream neurons compared to the other NTS populations that we have examined in this study.

The finding that silencing NTS$^{Calcr}$, NTS$^{Lepr}$, and NTS$^{Cck}$ cells increases food intake and body weight in HFD-fed animals, but barely (if at all) alters these parameters in chow-fed animals indicates the importance of these cells for the restraint of food intake specifically during the consumption of palatable calorically dense food. This might indicate that these NTS cell types play more prominent roles in the nutrient- (rather than volume/gut stretch) based suppression of feeding; alternatively, it might indicate that NTS neurons such as these play a stronger role in limiting hedonic overeating than in the restraint of physiologic need-based feeding. Disentangling these possibilities will require a substantial amount of future work.

Despite the prominent roles that NTS$^{Calcr}$, NTS$^{Lepr}$, and/or NTS$^{Cck}$ cells plays in the suppression of food intake and body weight in HFD-fed animals, silencing these NTS neurons failed to blunt the VSG-mediated decrease in food intake and body weight. Thus, other neural systems must mediate most of the weight-loss response to VSG. While the neurons that mediate this response to VSG might lie outside of the NTS (e.g. in the AP and/or hypothalamus), the finding that feeding (and feeding in the context of VSG) activate many non-LCK NTS neurons suggests that additional NTS populations restrain food intake and may play roles in the control of feeding.

Although many sets of NTS neurons have been implicated in the suppression of feeding, only recently have single cell RNA sequencing methods been employed to define populations of NTS neurons in an unbiased manner (*Cheng et al., 2022*; *Ludwig et al., 2021a*; *Dowsett et al., 2021*; *Ludwig et al., 2021b*). Unlike *Calcr*, *Lepr* marks multiple populations of NTS neurons, including one population (GLU2) that controls respiratory function in response to leptin/energy balance (*Ludwig et al., 2021a*; *Ludwig et al., 2021b*; *Do et al., 2020*).

Importantly, *Cck* expression distributes across several informatically defined NTS populations (including some NTS$^{Lepr}$ cells), and it remains unclear which *Cck*-expressing NTS snRNA-seq population(s) might mediate the avoidance-provoking suppression of food intake. Some NTS$^{Cck}$ neurons may map to bioinformatically defined cell populations that mediate responses other than the suppression of food intake.

Furthermore, *Calcr* and *Lepr* do not mark all neurons within their bioinformatically-defined populations (GLU11 and GLU13, respectively); the other neurons within these populations presumably mediate similar functions but are not altered by *Calcr*- or *Lepr*-directed manipulations, respectively.

Hence several sets of non-NTS$^{LCK}$ cells might contribute to the ongoing control of food intake in mice where the NTS$^{LCK}$ cells are silenced. These could include: the remaining unperturbed neurons

from GLU11 and/or GLU13; the presumably aversive NTS neuron population marked by *Cck* expression; and other populations of NTS neurons that mediate the avoidance provoking and/or non-avoidance provoking suppression of food intake, in addition to neuron populations that lie outside of the NTS (including in the AP and/or hypothalamus). Going forward, it will be important to identify and define roles for all the NTS cells that respond to feeding and play roles in the suppression of food intake, as well as to identify neurons in the AP and elsewhere that may contribute to physiologic energy balance. Similarly, it will be important to understand potential differences in regulation and downstream targets for each population of NTS neurons that modulates food intake.

# Materials and methods

## Key resources table

| Reagent type (species) or resource | Designation | Source or reference | Identifiers | Additional information |
|---|---|---|---|---|
| Strain, strain background (*mus musculus*) | Calcr^Cre; C57BL6/J | Jackson Lab Strain | #037028 | |
| Strain, strain background (*mus musculus*) | Lepr^Cre: C57Bl6/J | Jackson Lab Strain | #17527 | |
| Strain, strain background (*mus musculus*) | Cck^Cre; C57Bl6/J | Jackson Lab Strain | #012706 | |
| Antibody | anti-FOS (rabbit monoclonal) | Cell Signaling Technology | #2250 | IF: 1:1000 |
| Antibody | anti-GFP (chicken polyclonal) | Aves Laboratories | GFP1020 | IF: 1:1000 |
| Antibody | anti-dsRed (rabbit polyclonal) | Takara | 632496 | IF: 1:1000 |
| Recombinant DNA reagent | AAV^FLEX-hM3Dq | DOI: 10.1172/JCI46229 | | |
| Recombinant DNA reagent | AAV^FLEX-TetTox-GFP | DOI: 10.1038/nn.4574 | | |
| Recombinant DNA reagent | AAV^FLEX-hM4Di | DOI: 10.1172/JCI46229 | | |
| Recombinant DNA reagent | AAV^Cre-GFP | DOI: 10.1073/pnas.042678699 | | |
| Recombinant DNA reagent | AAV^GFP | DOI: 10.1073/pnas.042678699 | | |
| Chemical compound, drug | CNO | Tocris Bioscience | #4936 | |

## Animals

Mice were bred in our colony in the Unit for Laboratory Animal Medicine at the University of Michigan; these mice and the procedures performed were approved by the University of Michigan Committee on the Use and Care of Animals (Protocol# 00011066) and in accordance with Association for the Assessment and Approval of Laboratory Animal Care and National Institutes of Health guidelines. Mice were provided with food and water ad libitum (except as noted below) in temperature-controlled rooms on a 12 hr light-dark cycle. For all studies, animals were processed in the order of their ear tag number, which was randomly assigned at the time of tailing (before genotyping). ARRIVE guidelines were followed; animals were group-housed except for feeding and CTA studies.

We purchased male and female C57BL/6 mice for experiments and breeding from Jackson Laboratories. *Lepr^cre* and *Calcr^cre* mice have been described previously (*Patterson et al., 2011*; *Pan et al., 2018*) and were propagated by intercrossing homozygous mice of the same genotype. *Cck^cre* mice (Jax stock No.: 012706) for breeding were purchased from Jackson Labs (Bar Harbor, ME). Compound genotypes were generated by intercrossing the various cre genotype.

## Stereotaxic injections

AAV^FLEX-hM3Dq, AAV^FLEX-hM4Di, AAV^FLEX-TetTox-GFP, and control viruses (e.g. AAV^GFP and AAV^cre-GFP) (*Sternson and Roth, 2014*; *Carter et al., 2015*; *Armbruster et al., 2007*) were prepared by the University of Michigan Viral Vector Core. The AAVs used in the manuscript were all serotype AAV8.

For injection, following the induction of isoflurane anesthesia and placement in a stereotaxic frame, the skulls of adult mice were exposed. After the reference was determined, a guide cannula with a pipette injector was lowered into the injection coordinates (NTS: A/P, –0.2; M/L,±0.2; D/V, –0.2 from the obex) and 100 nL of virus was injected for each site using a picospritzer at a rate of 5–30 nL/min with pulses. Five minutes following injection, to allow for adequate dispersal and absorption of the virus, the injector was removed from the animal; the incision site was closed and glued. The mice

received prophylactic analgesics before and after surgery. The mice injected with AAV$^{Flex-hM3Dq}$, AAV$^{Flex-hM4Di}$, AAV$^{Flex-TetTox-GFP}$, or control viruses were allowed at least 1 week to recover from surgery before experimentation. AAV$^{FLEX-hM3Dq}$ was injected into mice as a control for mice injected with AAV$^{FLEX-TetTox}$.

## Phenotypic studies

Lepr$^{TetTox}$, Cck$^{TetTox}$, LC$^{TetTox}$, and LCK$^{TetTox}$ mice and their controls were monitored from the time of surgery for chow feeding. For stimulation studies, DREADD-expressing mice and their controls that were at least 3 weeks post-surgery were treated with saline or drug (CNO, 4936, Tocris) at the onset of dark cycle, and food intake was monitored over 4 hr. For chronic food intake and body weight changes in DREADD-expressing animals, mice were given saline for 2–3 days prior to injecting saline or drugs twice per day (approximately 5:30 PM and 8:00 AM) for 2 or 3 days, followed by saline injections for another 1 or 3 days to assess recovery from the treatment.

## Perfusion and immunohistochemistry

Mice were anesthetized with a lethal dose of pentobarbital and transcardially perfused with phosphate-buffered saline (PBS) followed by 10% buffered formalin. Brains were removed, placed in 10% buffered formalin overnight, and dehydrated in 30% sucrose for 1 week. With use of a freezing microtome (Leica, Buffalo Grove, IL), brains were cut into 30 μm sections. Immunofluorescent staining was performed using primary antibodies (FOS, #2250, Cell Signaling Technology, 1:1000; GFP, GFP1020, Aves Laboratories, 1:1000; dsRed, 632496, Takara, 1:1000), antibodies were reacted with species-specific Alexa Fluor-488,–568 or –647 conjugated secondary antibodies (Invitrogen, Thermo Fisher, 1:200). Images were collected on an Olympus (Center Valley, PA) BX53F microscope. Images were pseudocolored using Photoshop software (Adobe) or Image J (NIH).

## Conditioned taste avoidance (CTA)

Two types of CTA assay were conducted, as described previously (*Cheng et al., 2020a*): Saccharin CTA: Mice that were 8 weeks of age or older were individually housed in standard cages with low wire tops and free access to food and the lixits were removed. Mice were habituated to two water-containing bottles for 3–5 days until they learned to concentrate their daily water consumption into these two water bottles. On the conditioning day, the mice received only two saccharin (0.15%, 240931, Sigma) bottles. Following the 60 min exposure to saccharin, mice were injected intraperitoneally with the desired stimulus (vehicle control (0.9% NaCl), lithium chloride (0.3 M, 203637, Sigma)) in a volume equivalent to 1% of each animal's body weight (10 mL/g), or CNO (1 mg/kg, 4936, Tocris Bioscience) Access to the two saccharin bottles continued for an additional 1 hr, followed by the return of normal water bottles. Two days later, each mouse received access to two water bottles (one containing 0.15% saccharine, the other containing water), and the amount of fluid ingested from each water bottle was measured.

HFD CTA: following an overnight fast, mice were provided with one hour access of HFD (D012492, Research Diets), followed with the relevant stimuli, followed by an extra one hour of HFD access before returning of normal chow. On the post-conditioning day, fasted mice received access to both HFD and chow and the consumption of each was measured.

## Mouse vertical sleeve gastrectomy

Mice had ad libitum access to water and 60% lard-based HFD (Research Diet, New Brunswick, NJ; Cat. No. D12492) for 6–8 weeks prior to undergoing Sham or VSG surgery. Under isoflurane/O$_2$ mixture anesthesia, all mice received a midline incision in the ventral abdominal wall and the stomach was exposed. For VSG, approximately 80% of the stomach was transected along the greater curvature using an Echelon Flex Powered Vascular 35 mm Stapler Model (PVE35A; Ethicon endo-surgery) creating a gastric sleeve. The Sham surgery was performed by the application of gentle pressure on the stomach with blunt forceps for 15 s. All mice received one dose of Buprinex (0.1 mg/kg) and Meloxicam (0.5 mg/kg) immediately after surgery. Post-operatively, all mice received 1 ml warm saline subcutaneously on the first postoperative day and analgesic treatment with meloxicam (0.5 mg/kg) for 3 consecutive days. Animals were placed on DietGel Boost (ClearH$_2$O; Postland ME) for 2 days before and 3 days after surgery before pre-operative solid diet (60% HFD) was returned on day 4. Body weight and general health were observed daily for the first 10 days post-surgery. Food intake and

body weight were measured weekly for the duration of the study. Body composition was measured using nuclear magnetic resonance (EchoMRI–900, EchMRI LLC, Houston, USA) before surgery and periodically throughout the study.

### Glucose/insulin/acetaminophen assay

During glucose tolerance tests, all mice were fasted for 4–5 hr prior to oral administration of dextrose (2 g/kg body weight). Tail vein blood glucose levels were measured using Accu-Chek glucometers (Accu-Chek Aviva Plus, Roche Diagnostics) or Biosen C-line glucose analyzer (EKF diagnostic) at 0, 15, 30, 45, 60, 90, and 120 min post glucose administration. Gastric emptying rate was assessed by an oral gavage of glucose (2 g/kg) with acetaminophen (100 mg/kg) in 4–5 hr fasted mice. Blood was collected from the tail vein at baseline and 10 min after gavage in EDTA-coated microtubes. Plasma acetaminophen levels were used to assess the rate of gastric emptying and were measured using spectrophotometry assay (Sekisui Diagnostics). Plasma insulin levels were determined using ELISA colorimetric insulin assay kit (Crystal Chem).

### Metabolic chamber

A subset of mice was placed into an automated system equipped with metabolic chambers (TSE Systems International Group, Chesterfield, MO) to measure indices of energy homeostasis 5–7 week after surgery. Data were recorded continuously over the course of 7 days, and the last 4 days were used in the analysis.

### Statistics

Statistical analyses of the data were performed with Prism software (version 8). Two-way ANOVA, paired or unpaired t-tests were used as indicated in the text and figure legends. In detail, multiple groups with one variable comparison were performed using one-way ANOVA post hoc Tukey. Multiple groups with two variables comparison were performed using two-way ANOVA post hoc Sidak. Multiple group comparisons over time were performed using repeated measures ANOVA post hoc Tukey. All data are presented as Mean ± SEM. Two groups direct comparisons were performed using t-test. Two groups multiple comparisons were also performed using Multiple t-test with Bonferroni correction. Data were considered statistically significant when $p < 0.05$.

## Acknowledgements

We thank Paula Goforth, PhD, David Olson, MD, PhD, and Randy Seeley, PhD, as well as members of the Myers and Sandoval labs for helpful discussions. Special thanks to Stace Kernodle and Kelli Rule for superb technical support. Funding: Research support was provided by NIH P01 DK117821 (projects 1 and 3 to MGM and DS, respectively) and from the Michigan Diabetes Research Center (NIH P30 DK020572, including the Molecular Genetics and Animal Studies Cores), and the Marilyn H Vincent Foundation (MGM).

## Additional information

### Competing interests

Martin G Myers: receives research support from AstraZeneca and Novo Nordisk. Darleen Sandoval: is a consultant for Metis Therapeutics. The author declares that they have no other conflicts of interest. The other authors declare that no competing interests exist.

## Funding

| Funder | Grant reference number | Author |
|---|---|---|
| National Institute of Diabetes and Digestive and Kidney Diseases | P01 DK117821 | Darleen Sandoval |
| National Institute of Diabetes and Digestive and Kidney Diseases | P30 DK020572 | Martin G Myers |

The funders had no role in study design, data collection and interpretation, or the decision to submit the work for publication.

## Author contributions

Weiwei Qiu, Conceptualization, Resources, Formal analysis, Visualization, Methodology, Writing – original draft, Writing – review and editing; Chelsea R Hutch, Conceptualization, Data curation, Formal analysis, Investigation, Visualization, Writing – original draft, Writing – review and editing; Yi Wang, Investigation, Visualization, Writing – review and editing; Jennifer Wloszek, Rachel A Rucker, Investigation, Writing – review and editing; Martin G Myers, Darleen Sandoval, Conceptualization, Resources, Supervision, Funding acquisition, Writing – original draft, Writing – review and editing

## Author ORCIDs

Jennifer Wloszek ⓘ http://orcid.org/0000-0003-4914-8877
Rachel A Rucker ⓘ http://orcid.org/0000-0002-1434-9401
Martin G Myers ⓘ https://orcid.org/0000-0001-9468-2046
Darleen Sandoval ⓘ https://orcid.org/0000-0003-3669-3278

## Ethics

Mice were bred in our colony in the Unit for Laboratory Animal Medicine at the University of Michigan; these mice and the procedures performed were approved by the University of Michigan Committee on the Use and Care of Animals (Protocol#00011066) and in accordance with Association for the Assessment and Approval of Laboratory Animal Care and National Institutes of Health guidelines. Mice were provided with food and water ad libitum (except as noted below) in temperature-controlled rooms on a 12-hour light-dark cycle. For all studies, animals were processed in the order of their ear tag number, which was randomly assigned at the time of tailing (before genotyping). ARRIVE guidelines were followed; animals were group-housed except for feeding and CTA studies. All surgery was performed under isoflurane anesthesia and every effort was made to minimize suffering.

## Decision letter and Author response

Decision letter https://doi.org/10.7554/eLife.85640.sa1
Author response https://doi.org/10.7554/eLife.85640.sa2

# Additional files

## Supplementary files
• MDAR checklist

## Data availability
All data generated are included in the manuscript and supplemental figures.

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
