## [Editor Report]

This important study describes the role of several populations of NTS neurons in the control of energy balance. The authors provide solid evidence to show that simultaneously stimulating different populations in the NTS induces more potent effects on eating and weight. The work will be of interest to neuroscientists working on neural regulations of energy metabolism.

---

## [Decision Letter]

**Decision letter after peer review:**

Thank you for submitting your article "Multiple NTS Neuron Populations Synergistically Suppress Physiologic Food Intake but are Dispensable for the Response to VSG" for consideration by *eLife*. Your article has been reviewed by 2 peer reviewers, and the evaluation has been overseen by a Reviewing Editor and Mone Zaidi as the Senior Editor. The following individual involved in the review of your submission has agreed to reveal their identity: Thomas Lutz (Reviewer #1).

Essential revisions:

1) Please identify the released neurotransmitters upon activation of the respective neuronal populations and provide a clear hypothesis.

2) Please provide mechanistic insights into the phenomena examined as pointed out by Reviewer #2, such as deciphering the involved neural circuits in satiety or CTA, determining how the effects of activating Cck neurons override the satiety effects of activating Lepr and Calcr neurons, determining how activating combinations of neurons produces more robust effects on food intake than activating single populations, and/or defining the neurons promoting weight loss after VSG or least showing that such neurons exist in the NTS.

3) Please also address other comments /concerns brought up by the two reviewers.

*Reviewer #1 (Recommendations for the authors):*

I do have a few comments.

1) It would be interesting to know which neurotransmitters are released upon activation of the respective neuronal populations. Any idea?

2) Line 96ff: This is rather a summary of results than an introduction to the topic of research. Please revise. A clear hypothesis is missing.

3) Line 135: The term "additively" should be avoided. Additivity has not been tested because it would necessitate a combination of various concentrations/strengths of stimulation/inhibition.

4) The term "satiety" refers to specific controls of the inter-meal interval. This has not been tested here.

5) Line 225ff: In Figures 4G and 4H, two animals in the CNO group seem to be very different from the rest of the group. Are these the same animals in 4G and 4H, and what could be the reason for the different responses?

6) Some statements in the Discussion could be referenced better. E.g., line 299ff

7) Line 369ff: please describe for all experiments the nature of the control mice. Are these wild-type mice (non-litter mates) in most cases?

8) Line 407, but also throughout the text: the tests should be referred to as avoidance rather than aversion tests. Please correct throughout.

*Reviewer #2 (Recommendations for the authors):*

This paper would be more valuable if the authors used these findings as a foundation for experiments that address critical issues, for example:

1) Explore the neural circuits involved in satiety or CTA.

2) Determine how activating Cck neurons overrides the satiety effects of activating Lepr and Calcr neurons.

3) Determine how activating combinations of neurons produces more robust effects on food intake than activating single populations, e.g., does it provide more input to the same post-synaptic neurons or activate more post-synaptic targets?

4) Identify the neurons that do promote weight loss after VSG, or at least show that uch neurons exist in the NTS.

Editorial suggestions:

Line 2, The VSG abbreviation is not likely to be recognized by most readers.

Line 36, Why is the nucleus tractus solitarius in italics when other brain regions are not?

Line 115 and elsewhere including figures (.e.g, Supp 4), why is FOS in all capital letters, when other protein names are not?

Line 118, "As we showed previously" needs a reference.

Line 190, "cre-dependent AAV TetTox. " Cre should have a capital letter to be consistent with use elsewhere. The AAV is not Cre-dependent, it carries a Cre-dependent TetTox effector gene.

Lines 210 and 231, same suggestion as line 190.

Line 258, the hyphen should either be a comma or em dash.

Line 315, "calorically-dense" does not need a hyphen because calorically is an adverb.

Line 373, What was the original source of these AAV vectors?

Line 386, Is LepRb meant to be distinct from Lepr used on line 154?

Line 504, JCI is not an acceptable journal abbreviation.

Line 519, 537, and 546 Journal names should be abbreviated.

Figure 1A, Adding Bregma levels to all histology figures would be useful.

Figure 1 B and C, Ordinate would look better with half as many numbers.

Figure 5D, “TotTox” should be TetTox.

Figure 6, C, D and E, "Body composition" does not seem like the right label and it is not measured in grams.

---

## [Author Response]

Essential revisions:1) Please identify the released neurotransmitters upon activation of the respective neuronal populations and provide a clear hypothesis.

We have now enumerated the neurotransmitters found in each of the NTS neuron populations (introduction, page 4- end of first paragraph).

2) Please provide mechanistic insights into the phenomena examined as pointed out by Reviewer #2, such as deciphering the involved neural circuits in satiety or CTA, determining how the effects of activating Cck neurons override the satiety effects of activating Lepr and Calcr neurons, determining how activating combinations of neurons produces more robust effects on food intake than activating single populations, and/or defining the neurons promoting weight loss after VSG or least showing that such neurons exist in the NTS.

We have now added data demonstrating differences in the activation of FOS-IR in the downstream targets of our NTS neuron types, alone or in combination (new Figure 6). Our findings reveal that each population (NTS^Lepr^, NTS^Calcr^, and NTS^Cck^) activates an at least partially distinct set of neurons and that only NTS^Cck^ cells activate the known aversive PBN CGRP cells. These data suggest that the cumulative effects mediated by each of these NTS populations stem in part from their ability to activate at least partly distinct populations of downstream neurons. Unfortunately, it is outside of the scope of this manuscript (and the realm of the possible) to define the neurons that mediate the response to VSG, and we have now reorganized the manuscript to clarify that our VSG data serve to reveal that additional populations of neurons (other than NTS^LCK^ cells) must contribute to the restraint of feeding.

3) Please also address other comments /concerns brought up by the two reviewers.

We have addressed the other comments as detailed below.

Reviewer #1 (Recommendations for the authors):I do have a few comments.1) It would be interesting to know which neurotransmitters are released upon activation of the respective neuronal populations. Any idea?

We have now enumerated the neurotransmitters found in each of the NTS neuron populations (introduction, page 4- end of first paragraph).

2) Line 96ff: This is rather a summary of results than an introduction to the topic of research. Please revise. A clear hypothesis is missing.

We have now revised this portion of the Introduction, including by stating a hypothesis (bottom of Page 4).

3) Line 135: The term "additively" should be avoided. Additivity has not been tested because it would necessitate a combination of various concentrations/strengths of stimulation/inhibition.

We have now removed this term throughout.

4) The term "satiety" refers to specific controls of the inter-meal interval. This has not been tested here.

We have now omitted this term throughout.

5) Line 225ff: In Figures 4G and 4H, two animals in the CNO group seem to be very different from the rest of the group. Are these the same animals in 4G and 4H, and what could be the reason for the different responses?

One of the animals is the same; the other is not. Regarding the reasons underlying the lack of response, occasionally mice drink little fluid or eat little HFD during the conditioning phase, which can interfere with association formation.

6) Some statements in the Discussion could be referenced better. E.g., line 299ff

We have now added references to the Discussion, including the referenced line.

7) Line 369ff: please describe for all experiments the nature of the control mice. Are these wild-type mice (non-litter mates) in most cases?

We have now described the nature of the control mice used in each experiment in the Figure Legend appurtenant to the experiment.

8) Line 407, but also throughout the text: the tests should be referred to as avoidance rather than aversion tests. Please correct throughout.

We have made this change.

Reviewer #2 (Recommendations for the authors):Editorial suggestions:Line 2, The VSG abbreviation is not likely to be recognized by most readers.

We have now omitted the reference to VSG from the title.

Line 36, Why is the nucleus tractus solitarius in italics when other brain regions are not?

*Nucleus tractus solitarius* is Latin.

Line 115 and elsewhere including figures (.e.g, Supp 4), why is FOS in all capital letters, when other protein names are not?

All protein names are capitalized, in accordance with convention (see, for instance, enumeration of receptors, bottom of page 4). Names of genes and transcripts are shown in italics, with only the first letter capitalized (also in accordance with convention).

Line 118, "As we showed previously" needs a reference.

We have provided the requested reference.

Line 190, "cre-dependent AAV TetTox. " Cre should have a capital letter to be consistent with use elsewhere. The AAV is not Cre-dependent, it carries a Cre-dependent TetTox effector gene.Lines 210 and 231, same suggestion as line 190.

We corrected this error throughout.

Line 258, the hyphen should either be a comma or em dash.

We corrected this error.

Line 315, "calorically-dense" does not need a hyphen because calorically is an adverb.

We corrected this error.

Line 373, What was the original source of these AAV vectors?

We have now provided references for the AAV vectors.

Line 386, Is LepRb meant to be distinct from Lepr used on line 154?

LepRb refers to the specific isoform of the protein; *Lepr* is the gene/transcript.

Line 504, JCI is not an acceptable journal abbreviation.

The journal name is JCI Insight.

Line 519, 537, and 546 Journal names should be abbreviated.

Corrected.

Figure 1A, Adding Bregma levels to all histology figures would be useful.

We have added this information.

Figure 1 B and C, Ordinate would look better with half as many numbers.

We have amended the panels as requested.

Figure 5D, “TotTox” should be TetTox.

We have corrected this error.

Figure 6, C, D and E, "Body composition" does not seem like the right label and it is not measured in grams.

We have corrected the label to read “Tissue Mass (g)”.